# Effect of a Nanocellulose Addition on the Mechanical Properties of Paper

**DOI:** 10.3390/polym16010073

**Published:** 2023-12-26

**Authors:** Josef Bárta, Kateřina Hájková, Adam Sikora, Tereza Jurczyková, Daniela Popelková, Petr Kalous

**Affiliations:** 1Department of Wood Processing and Biomaterials, Faculty of Forestry and Wood Sciences, Czech University of Life Sciences Prague, Kamýcká 129, 165 00 Prague, Czech Republic; bartajosef@fld.czu.cz (J.B.); sikoraa@fld.czu.cz (A.S.); jurczykova@fld.czu.cz (T.J.); xkalp019@studenti.czu.cz (P.K.); 2Faculty of Forestry and Wood Sciences, Czech University of Life Sciences Prague, Kamýcká 129, 165 00 Prague, Czech Republic; popelkovad@fld.czu.cz

**Keywords:** nanocellulose, soda pulp, kraft pulp, tensile strength

## Abstract

Nowadays, the emphasis is on increasing the durability of all products. For this reason, it is also advisable to look into extending the durability of paper products. The main reason for using flax pulp is that flax and cotton pulp are widely used for the production of banknotes due to their higher strength. This paper deals with flax pulp with the addition of nanocellulose, which should further enhance the mechanical properties of the pulp. The tensile strength, breaking length, and tensile energy absorption index were evaluated as the key mechanical properties. At the same time, the effect of the addition of nanocellulose, whether it was added to the pulp mass or applied to the later produced paper as a spray or coating, was tested in comparison to paper without the addition of nanocellulose. The best mechanical properties, i.e., tensile strength, were achieved for the highest addition of 5% of nanocellulose into the pulp, at 24.3 Nm∙g^−1^, and for the coating application, at 28.7 Nm∙g^−1^, compared to the flax pulp without the addition, where the tensile strength was 20.5 Nm∙g^−1^. The results of this research are used for the assessment of nanocellulose as a natural compatible additive to enhance the strength properties of cellulose-based materials.

## 1. Introduction

As an undeniably indispensable everyday material, paper is the target of interest in many studies. Those studies usually assess the enhancement to its strength and other mechanical properties. Das et al. [1] investigated the effect of nanocellulose as a nanofiller on the strength properties after using an inorganic filler. The strength and optical properties of the paper in the wet and dry parts were improved by the surface treatment using cellulose nanocrystals (CNCs) and cellulose nanofibers (CNFs), respectively [2]. Nanocellulose was used as an additive in paper pulp by Li et al. [3], who focused mainly on specialty papers made from recycled fibers, where nanocellulose represented a great added value. Better mechanical properties of the paper and polymer films were also achieved using nanocellulose produced by enzymatic hydrolysis [4]. In addition to nanocellulose fibers made from cotton, nanofibrillated cellulose from kelempayan was extracted and applied to paper pulp. However, adding up to 10% nanofibrillated cellulose was required to increase the mechanical properties significantly [5]. Dinesh et al. [6] focused on producing nanocellulose from wood and nonwood fibers, and their mechanical properties improved when producing transparent paper. In addition to improving the mechanical properties by adding nanocellulose, antibacterial effects were demonstrated, allowing for the development of new degradable packaging [7].

For those purposes, nanocellulose derivates had adequate characteristics, which could improve some of the surface, mechanical, or optical properties. The application of nanocellulose in industry promises not only an improvement in the mechanical properties of the final product, but also something more essential for today’s demand—reducing the impact on the environment through renewable resources, sometimes called industrial or agricultural waste. Hence, nanocellulose can significantly contribute to the paper industry as an environmentally friendly coating additive. This fact opens the gates for nanocellulose utilization. Nowadays, many commonly used paper coating materials, such as polymers or waxes [8], have a fossil origin.

Nanocellulose meets today’s demand to contribute to a circular economy, and its production and use can help industries move from a linear economy to a circular one [9]. It is a material that can be produced from many lignocellulosic materials, such as wood waste, plant or other agricultural waste, etc. Each lignocellulosic source could produce nanocellulose with different properties and characteristics, thus bringing many ways to assess the different properties of nanocellulose and its type of production and utilization [10]. Some studies have examined different lignocellulosic sources for producing nanocellulose [11,12,13]. The structure and properties of this material are sufficient for production, and the product itself in the form of nanocellulose can be readded to the primary fibers of the pulp and paper, whose properties it can theoretically improve. For these reasons, a large amount of recent research has discussed the production and utilization of nanocellulose [14,15,16]. A significant benefit of nanocellulose is its biodegradability, one of the main characteristics many additives lack [17]. Nanocellulose has a wide range of applications. It could be used in many industry sectors, such as paper, food, medicine, textile, etc.

This paper focuses on the utilization of nanocellulose in the paper industry. Some studies have carried out the application and improvement of the properties of paper with nanocellulose. It has been found that many paper properties, such as the tensile strength, relative elongation, and other mechanical properties, can be improved by applying nanocellulose [17]. Nanocellulose can be added to the paper in different ways: into batches during the papermaking process or as a coating on the final paper product. Deng et al. [11] and Barbash et al. [18] published that applying nanocellulose on the surface can improve the barrier properties of the paper as well as the mechanical properties, such as the burst index or tensile index [19].

This research aims to deepen the knowledge about using nanocellulose in paper production. The effect of applying nanocellulose into the pulp mass at different concentrations and to the final paper by spraying and coating was investigated. This effect was evaluated by measuring the mechanical properties of the produced paper compared to paper without the nanocellulose application.

## 2. Materials and Methods

### 2.1. Materials

Kraft pulp from conifers from Mondi Štětí (Štětí, Czech Republic) was used for the preliminary experiments. The kraft unbleached pulp was characterized by a delignification degree of 24.9 and 49.7, and the bleached pulp reached a Kappa number of 19.2. A soda flax pulp with a Kappa number of 19.9 from the Delfort Group (Olšany, Czech Republic) was used for further measurements.

To increase the strength, nanocellulose was added as nanofibrils, both the CNF–Slurry–SMC nanofibril type and the TEMPO-oxided nanofibril type, which were supplied by CelluloseLab (Fredericton, NB, Canada). Both types are produced initially from softwood kraft pulp, hardwood kraft pulp, kenaf pulp, and cotton pulp. Nanofibrils were prepared by the nanofibrillation of cellulose using a high-pressure homogenizer or a supermassive colloid (SMC). Other properties and essential characteristics of the nanocellulose reported by the manufacturer are listed in Table 1.

### 2.2. Application of the Cellulose Nanofibrils

The application of the nanocellulose was implemented in two ways. The first method was the incorporation of the nanocellulose into the pulp, i.e., into the pulp during paper production (0.5% NC, 1% NC, 3% NC, and 5% NC). Because there is a risk that the nanofibers will leak into the subsieve water with this application, the method of the final surface treatments was also chosen—spraying and coating in a single layer and two layers (one coating and two coatings).

The test handsheets were produced, so the first series was prepared with 0.5, 1, 3, and 5% of the CNF–Slurry–SMC nanocellulose, and the second only from the pulp without the nanocellulose addition. The pulp, including and without the nanocellulose, was weighed so that the resulting handsheet had a basis weight of approximately 70–80 g∙m^−2^. The pulp thus weighed was pulped using a Lorentzen and Wettres (Stockholm, Sweden) laboratory pulper and subsequently transferred to a RAPID-KÖTHEN RK-2A (Birkenau, Germany) laboratory sheeting machine, all according to ISO 5269-2:1998 [20].

The handsheets without the nanocellulose addition into mass were subjected to the application of TEMPO-oxided nanofibrils as a 0.5–3% fiber concentration solution to the material by spraying and coating. In the case of spraying, we applied 2.5 g of TEMPO-oxided nanofibers to the laboratory handsheets, and in the case of coating, we applied 3.5 g for one coating and 4.5 g for two coatings. At the same time, laboratory sheets without the addition of nanocellulose were maintained as reference samples. The spraying method was performed by introducing the NC solution into a laboratory liquid spraying device. Approximately equal amounts of the NC solution were applied to the surface of the laboratory handsheets. This method was intended to simulate the spraying that would be possible in a paper machine. The coating was applied evenly with a brush to the entire surface of the laboratory handsheets. This was followed by the absorption of the coating into the paper pulp; the coatings were done in one or two layers.

### 2.3. Mechanical Properties

Before measuring the mechanical properties, the laboratory handsheets were first conditioned according to TAPPI T 402 sp-08 [21]. The produced sheets were tested for tensile properties according to ČSN EN ISO 1924-2 [22] on FRANK-PTI equipment (Birkenau, Germany). Measured mechanical properties include the breaking length, relative elongation, tensile index, absorption tensile work, and tensile work absorption index. In addition, the burst index was also analyzed for the pre-experiments according to ISO 2758:2014 [23] using equipment from FRANK-PTI (Birkenau, Germany). All these strength parameters were determined on at least ten samples, and the results were statistically processed. In addition to the strength tests, the air permeability was determined according to the Gurley method of ISO 5636-5:2013 [24] using equipment from Lorentzen and Wettres (Stockholm, Sweden).

### 2.4. Duncan’s Test

Duncan’s test as a statistical post hoc method usually follows the analysis of variance, ANOVA. It describes the specific statistical means of averages that differ from each other in a data group. After the ANOVA analysis, which confirmed the statistical differences between means, the Duncan test compared the individuals in detail due to multiple pairwise comparisons among the groups to confirm and describe in more detail which data groups were significantly different [25].

### 2.5. Scanning Electron Microscopy (SEM)

Samples were mounted on specimen stubs, sputter-coated with gold in the Sputter Coater JFC-1300 (JEOL, Tokyo, Japan) under argon atmosphere, and examined by scanning electron microscopy using a JEOL JSM-IT500HR instrument (JEOL, Tokyo, Japan) operating at 15 kV.

## 3. Results and Discussion

### 3.1. Pre-Experiment with Kraft Pulp

The pre-experiment was designed to investigate the effect of adding nanocellulose to kraft softwood pulp, which was bleached or unbleached. The addition of CNF–Slurry–SMC nanocellulose into the mass was 0.5, 1, and 3%. The effect of each nanocellulose addition on the strength properties of the final laboratory handsheets is shown in Table 2.

According to the measured values from the pre-experiment, it was decided to use the bleached pulp for the next experiment, which was cooked to a low Kappa number, 19.9. A lower Kappa number implies a lower lignin content. Thus, a higher cellulose content is more advantageous for the application of NC in terms of the free -OH groups with which the NC react and bind. This soda technology produces flax pulp. The flax was used as the raw material because of its cotton-like properties. Since the burst index showed an increase even with a slight addition of nanocellulose, as well as the effect on the air permeability, the effect of the nanocellulose addition on tensile properties was evaluated as a priority for the next experiment.

### 3.2. Experiment with Soda Pulp

For the main experiment, a soda flax pulp was used, into which the CNF–Slurry–SMC nanocellulose was applied into the mass in amounts of 0, 0.5, 1, 3, and 5%, and in the form of TEMPO-oxided nanofibrils by spraying and coating on finished laboratory handsheets. The flax was also chosen because of the high strength properties of the original material, as reported by Bampach [26]. One series of handsheets without the addition of nanocellulose was kept for comparison. Before measuring the mechanical properties, the handsheets were first conditioned. The results of the tensile properties are presented in graphs in Figure 1 and Table 3 and Table 4 in the form of an analysis of homogeneous groups using Duncan’s test.

Figure 1a shows the results for the breaking length. This figure is graphically similar to Figure 1c, with results for the tensile index. These results suggest that these test scores are correlated. The breaking length after the addition of 0.5% NC into the mass was about 2.09 km, i.e., a 68.5% increase compared to 1.24 km for the reference sample. Moreover, similar results in comparison with the tensile index were observed for the 1%, 3%, and 5% addition of NC, which are described in more details in the next paragraph. The results associated with applying NC in the amount of 5% appear to be the most appropriate in improving the tensile index. However, the addition of 0.5% will increase the tensile index sufficiently. A 5% addition of NC into the pulp mass provides a tensile index of 2.88 Nm∙g^−1^ (a 132.3% increase). For the spray applications, the results are similar to the 5% NC use. There was an increase in the breaking length to 2.79 km (125% increase). For the coating applications, one coating had an increase in the breaking length up to 2.93 km (a 136.3% increase), and two coatings had an increase in the breaking length up to 2.92 km (a 135.5% increase). There was no significant increase in this coating application compared to the direct application of NC at 5%. When comparing the number of coatings, one coating seemed to be sufficient for the required increase in the mechanical properties, because the second coating no longer significantly affected these properties, as can be seen from the results. This is also evident from the work of Lavoine et al. [27], where they state that further application of the material did not change the breaking length; on the contrary, in some samples, they achieved even worse results than with a single application of microfibrillated cellulose.

According to another study [28], the tensile strength could be increased from around 15 Nm∙g^−1^ to about 40 Nm∙g^−1^ by using up to 9% CMFs (cellulose microfibrils). In this research, the tensile index of the reference sample was about 12.1 Nm∙g^−1^. The addition of 0.5% of NC increased this parameter to 20.5 Nm∙g^−1^ (a 69.4% increase). There was still an increase of 1% with the usage of NC compared to the reference, but also a slight decrease compared to a 0.5% usage of NC. Adding 3% of NC increased the tensile index to 20.7 Nm∙g^−1^ (71.1% increase) compared to the reference. The highest increase up to 28.2 Nm∙g^−1^ (a 133.1% increase) was observed for the sample with a 5% dosage of NC into the pulp mass. For this sample, the highest value of the breaking length was also measured. These increases may be due to the laboratory handsheets having randomly oriented fibers; randomly oriented nanocellulose was added to them, which will ultimately affect the final tensile strength.

Some studies mention that the relative elongation with the tensile strength in paper/film evaluation is usually used to describe the maximum strength and flexibility of paper/film [29]. Another study also confirmed that using CNPs (cellulose nanoparticles) in amount between 1 and 5% in the film composed of PVA (polyvinyl alcohol biopolymer) + CNPs + fennel seed oil increased the tensile strength and relative elongation with an increasing CNP concentration in that film. However, this study describes that the film composition of corn starch + NC + glycerol, and other film compositions of sugar palm starch + SPNFCs + glycerol + sorbitol, decreased the relative elongation [30]. The study published by Li et al. [17] confirms that the specific percentage use of NC could increase the tensile strength by 50.3% and the relative elongation by 26.3%. From the results of this research paper, the relative elongation for NC use in papermaking had the highest increase at a 5% NC use. The results for the 5% of the usage of NC indicate a relative elongation of 1.88% compared to 1.54% for the reference samples (22.1% increase). The second-best result of the relative elongation, i.e., 1.80% (a 16.9% increase), was determined for the samples with the addition of the 1% NC. Spraying application results for the relative elongation are worse than for the samples where 1, 3, and 5% NC were dosed into the pulp mass. This could be due to the accidental application of nanocellulose colloid. This sample series’ average relative elongation value was 1.52%, which means a 1.3% decrease compared to the reference sample. With coatings, better results in terms of the relative elongation were obtained, i.e., 2.44% (a 58.4% increase) for one coating and 2.72% (a 76.6% increase) for two coatings of the NC application. Coatings seem to provide the highest value of the relative elongation for all examined series, but there is also a higher variability in the results.

The tensile work absorption index is related to the next part of the trial testing. When using 0.5% NC in the mass application, there was an increase up to 0.23 J∙g^−1^ (a 15% increase) compared to the reference samples with a value of approximately 0.20 J∙g^−1^. For the 1 and 3% NC use, the increase was approximately the same for the 1% (0.26 J∙g^−1^, i.e., a 30% increase) and for 3% (0.26 J∙g^−1^, i.e., a 30% increase). The 5% NC usage in a mass application had the highest increase in the tensile work absorption index of 0.32 J∙g^−1^ (a 60% increase). The application by spraying provided approximately the same values compared to the 5% NC usage into the mass (0.32 J∙g^−1^, i.e., a 60% increase). For the coatings, the increase in this parameter was the highest compared to all types of NC applications. For one coating, the tensile work absorption index was 0.33 J∙g^−1^ (a 65% increase), and for two coatings, the method was approximately 0.34 J∙g^−1^ (a 70% increase). Again, however, the values obtained for the coatings had a higher variability.

Table 3 shows the analysis of the homogenous groups based on the mean values of ε and BL. This test was made with a 95% confidence interval. For the results of ε, it can be said that the addition of nanocellulose during the paper production and spraying is classified in a homogeneous group, with insignificant statistical differences corresponding to the reference paper based on the comparison of the significance levels. Applying nanocellulose by both application coating methods created a higher homogeneous group with statistically significantly higher values than the reference group. In the case of BL, the effect of adding nanocellulose was more evident based on the statistical evaluation. The reference paper formed a separate homogenous group with the lowest average value for this characteristic. The second homogenous group, which had statistically significantly higher values than the reference, consisted of papers with 0.5%, 1%, and 3% additions of nanocellulose in the production process. The most statistically significant increase was achieved in the homogeneous group with the paper with a 5% addition of nanocellulose with the application of nanocellulose by spraying and both methods of coating the nanocellulose on the paper. The main objective of this study was to evaluate the mechanical properties of the papers with and without the addition of nanocellulose.

Regarding the breaking length, the result obtained was 2.8–2.9 km. Compared to the values in the work of Rahman and Uddin [31], the result for the untreated cotton pulp with nanocellulose alone was 2.5 km. In another work, they attempted to increase the breaking length by increasing the lignin and Caro’s acid content. They achieved values around 3.5 km by adding 6% Caro’s acid to the bagasse pulp [32]. Similar properties were also achieved for rice straw when the paper was made by biorefining with the addition of 90% alkali, with 3.0 km [33].

Table 4 shows the analysis of the homogenous groups based on the mean values of TI and TEAI. In the case of the TI characteristic, based on the statistical evaluation, identical homogeneous groups were formed as in the case of the BL characteristic shown in Table 3. On the other hand, in the case of TEAI, more homogeneous groups were formed based on the statistical evaluation using significance levels than in the previous case. The specific difference was that the paper with a 1% addition of nanocellulose was classified into two homogeneous groups based on the comparison of the significance levels. However, even here, a more significant statistical difference compared to the reference was achieved by the homogeneous group with the paper with a 5% addition of nanocellulose with the application of the nanocellulose by spraying, and both methods of coating the nanocellulose on the paper.

The tensile strength was compared with other publications. We were able to increase the tensile strength index to about 28 N·mg^−1^. After adding 0.9% nanocellulose to the cotton fibers, the tensile strength increased from the initial 42.9 N·mg^−1^ without the nanocellulose to 57.2 N·mg^−1^, which was as high an increase when using nanocellulose as in our case [34]. The authors [35] attempted to modify rice pulp by the sodium–ethanol process, but obtained a significant decrease in the mechanical strength relative to the reference sample from the original 58.66 N·mg^−1^; the value dropped to 11.15 N·mg^−1^.

However, compared to the wood pulp, consistently low results, even after adding the nanocellulose, were achieved for poplar, at 42.7 N·mg^−1^, and for willow, at 46.3 N·mg^−1^ [36]. For the pine pulp, it was around 55 N·mg^−1^; however, the authors achieved a similar result when 10% wheat was added to the pine pulp, with a result of 56 N·mg^−1^ [37]. Pulp from the wheat straw alone achieved results similar to the cotton, at 43.5 N·mg^−1^ [38].

It can also be observed that the additions of less than 3% did not significantly affect the resulting properties, as the nanocellulose fibrils were introduced into the subsieve waters of the paper machine.

In Figure 2 and Figure 3, it is possible to see the correlations between the selected characteristics. Based on the obtained results, the correlations of the selected characteristics with a 95% confidence interval were performed. Figure 2 shows the correlation between the relative elongation and the tensile index. Based on the absolute correlation r = 0.59592, this is a moderate positive dependence, i.e., as the relative elongation increases, the tensile index also increases with a moderate positive dependence.

Higher absolute values of the correlation index (r = 0.90874) were achieved for the correlation between the tensile work absorption index and the tensile index. In this case, it was a very strong positive correlation, i.e., when the tensile work absorption index increases, the tensile index will also increase with a very strong dependence.

Scanning electron microscope (SEM) micrographs are shown in Figure 4. The left side shows the samples without the addition of nanocellulose, and the right side shows the samples with the 5% addition of nanocellulose into the mass. In terms of the resolution, the samples located on top are magnified 200× and the samples located on the bottom are magnified 5.500×. In the sample with the addition of 5% nanocellulose, the nanofibers are clearly visible—small bright objects on the surface of the paper fibers. As seen in Figure 4, after the addition of the nanocellulose, the surface of the fiber was covered. Thus, less porosity can be observed. Based on this figure, it seems that the nanocellulose treatment can fill the vacant gaps between the fibers.

## 4. Conclusions

It is possible to observe the effect of the addition of the nanocellulose on the mechanical properties of the paper.

-Concerning the amount of cellulose added, the modification by spraying nanocellulose on the paper appears to be the most effective. At a 3 and 5% NC content, the properties are comparable to the papers using more NC, but applied differently.-In general, it can also be observed that additions of less than 3% do not significantly affect the resulting properties, as apparently the nanocellulose fibrils are introduced into the subsieve waters of the paper machine.-When NC is added during the actual production of the paper, a positive dependence on the amount of NC added can be observed.-The application of NC using a one-layer coating is practical, even at a lower content (4.8%) of the applied NC, compared to a two-layer coating of NC (6.8%). In contrast, the coating application is not new in production. Thus, it would not be difficult to introduce this application into the production process.-In the future, it would be good to look into the composition of nanocellulose or its production, as, recently, nanofibers containing lignin in addition to cellulose have been produced, which could have a positive impact on increasing the mechanical properties.

## Figures and Tables

**Figure 1 polymers-16-00073-f001:**
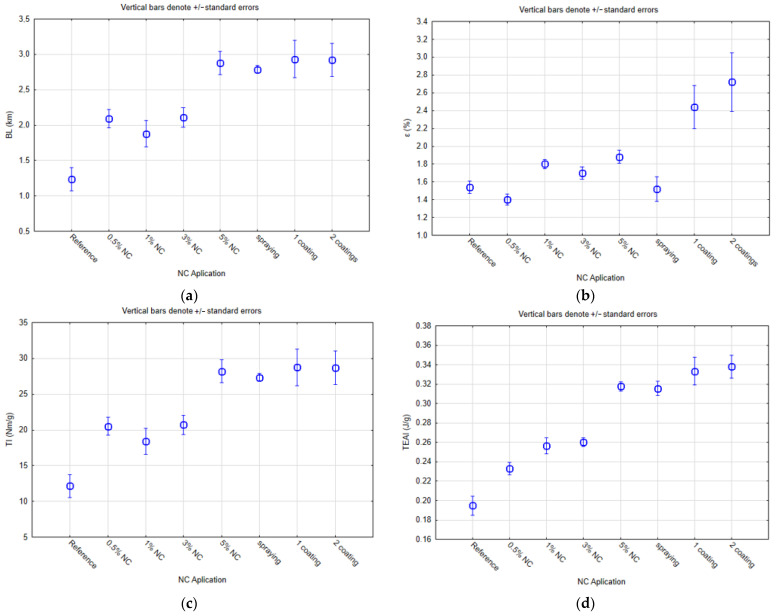
Tensile properties of soda flax pulp handsheets with nanocellulose ((**a**)—breaking length; (**b**)—relative elongation; (**c**)—tensile index; (**d**)—tensile work absorption index).

**Figure 2 polymers-16-00073-f002:**
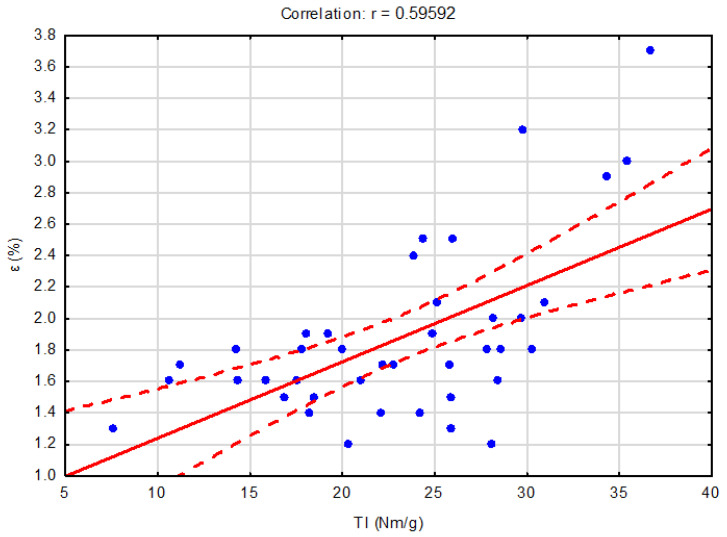
Dependence of the relative elongation on the tensile index.

**Figure 3 polymers-16-00073-f003:**
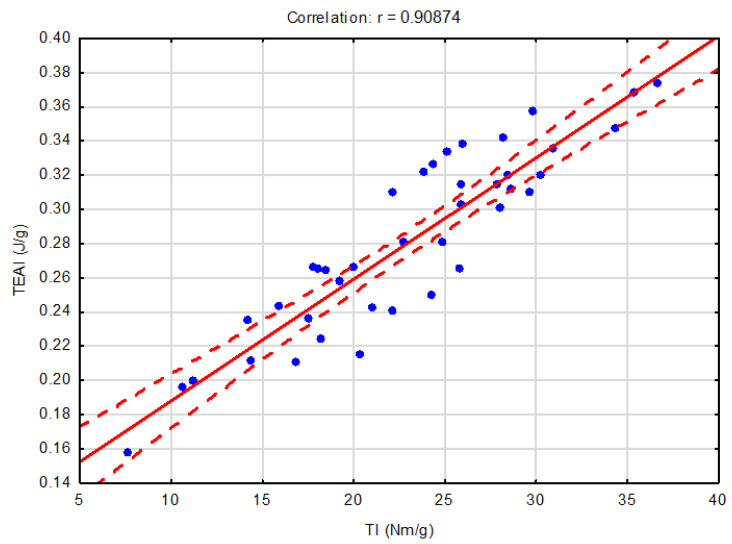
Dependence of the tensile work absorption index on the tensile index.

**Figure 4 polymers-16-00073-f004:**
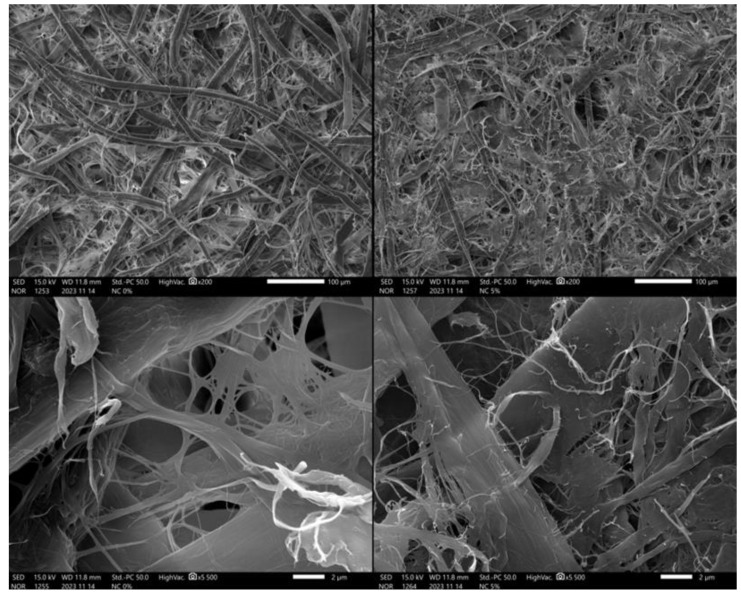
Scanning electron microscope (SEM) micrographs.

**Table 1 polymers-16-00073-t001:** Characteristics of the cellulose nanofibrils.

Type of Nanofibrils	Description	Dimensions		Group	Forms
		Width	Length		
CNF–Slurry–SMC	Cellulose nanofibrils prepared using supermassive colloid	30–80 nm	up to several hundred μm	hydroxyl	1–20% solids in the solution
TEMPO-Oxided	Cellulose nanofibril slurry	20–50 nm	0.5–80 μm	carboxyl, hydroxyl	0.5–3% solids in the solution

**Table 2 polymers-16-00073-t002:** Strength properties for the kraft pulp with the added nanocellulose.

Materials/Strength Properties	Amount of Nanocellulose	BI, kPa	G, s	BL, km	ε, %	TI, Nm∙g^−1^	TEAI, J∙g^−1^
Kraft bleached pulp, κ = 19.2	0%	69.48 (5.16)	0.58 (0.04)	0.85 (0.03)	1.29 (0.07)	8.37 (0.25)	0.08 (0.01)
	0.5%	80.98 (2.96)	0.62 (0.04)	1.03 (0.04)	1.50 (0.20)	10.07 (0.40)	0.11 (0.02)
	1%	82.32 (5.25)	0.60 (0.00)	0.93 (0.06)	1.57 (0.17)	9.14 (0.60)	0.11 (0.02)
	3%	101.30 (3.72)	0.52 (0.04)	0.86 (0.03)	2.88 (0.19)	8.48 (0.28)	0.19 (0.02)
Kraft unbleached pulp, κ = 24.9	0%	175.86 (2.72)	1.06 (0.05)	2.71 (0.10)	2.40 (0.12)	26.57 (1.03)	0.44 (0.03)
	0.5%	183.08 (7.05)	1.02 (0.04)	2.45 (0.15)	2.62 (0.17)	25.58 (1.36)	0.46 (0.06)
	1%	219.06 (8.51)	1.92 (0.11)	3.05 (0.23)	2.86 (0.21)	29.71 (2.19)	0.61 (0.09)
	3%	301.32 (8.57)	6.26 (0.05)	4.41 (0.12)	3.28 (0.13)	42.87 (1.10)	0.95 (0.08)
Kraft unbleached pulp, κ = 49.7	0%	35.80 (1.54)	0.36 (0.05)	1.85 (0.14)	1.34 (0.15)	11.68 (0.92)	0.07 (0.03)
	0.5%	49.90 (0.94)	0.50 (0.07)	2.03 (0.02)	1.43 (0.12)	19.96 (0.19)	0.20 (0.02)
	1%	102.38 (7.72)	1.22 (0.37)	2.37 (0.05)	1.64 (0.10)	23.53 (0.30)	0.02 (0.01)
	3%	168.60 (12.25)	2.80 (0.14)	2.78 (0.11)	2.13 (0.15)	27.23 (0.98)	0.41 (0.05)

κ—Kappa number; BI—burst index; G—air permeability (Gurley); BL—breaking length; ε—relative elongation; TI—tensile index; TEAI—tensile work absorption index. Standard deviation values are in parentheses.

**Table 3 polymers-16-00073-t003:** Analysis of the homogeneous groups using Duncan’s test; variable relative elongation and breaking length.

Application	ε, % Mean	1st Group	2nd Group	BL, km Mean	1st Group	2nd Group	3rd Group
Reference	1.540	****		1.236			****
0.5% NC into the mass	1.400	****		2.092		****	
1% NC into the mass	1.800	****		1.878		****	
3% NC into the mass	1.700	****		2.110		****	
5% NC into the mass	1.880	****		2.876	****		
Spraying (3.5% NC)	1.520	****		2.786	****		
Coating—1 layer (4.8% NC)	2.440		****	2.932	****		
Coating—2 layers (6.8% NC)	2.720		****	2.924	****		

**Table 4 polymers-16-00073-t004:** Analysis of the homogeneous groups using Duncan’s test; variable tensile index and tensile work absorption index.

Application	TI, Nm·g^−1^ Mean	1st Group	2nd Group	3rd Group	TEAI, J·g^−1^ Mean	1st Group	2nd Group	3rd Group	4th Group
Reference	12.144			****	0.195				****
0.5% nanocellulose	20.505		****		0.233			****	
1% nanocellulose	18.423		****		0.256		****	****	
3% nanocellulose	20.701		****		0.260		****		
5% nanocellulose	28.201	****			0.318	****			
Spraying (3.5% NC)	27.326	****			0.316	****			
Coating—1 layer (4.8% NC)	28.750	****			0.333	****			
Coating—2 layers (6.8% NC)	28.681	****			0.338	****			

## Data Availability

Data are available on request due to ethical restrictions. The data presented in this study are available on request from the corresponding author. The data are not publicly available due to unfinished research.

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
