# Peer review of "Effect of a Nanocellulose Addition on the Mechanical Properties of Paper"

_polymers, 2023, doi:10.3390/polym16010073_

Round 1

Reviewer 1 Report

Comments and Suggestions for Authors

The experimental article “Effect of nanocellulose addition on the mechanical properties of paper” fully corresponds to the profile of the Polymers publication. The topic of the article is very relevant, since the demand for paper, in particular and high-strength paper, is increasing many times due to the need to reduce the adverse impact on the environment. The material of the article is presented sequentially; in the “Results and Discussion” part, the authors compare their results with the data of other researchers in sufficient detail. However, there are a number of comments, after elimination of which the article can be published.

Notes:

1) Line 29: indicate literature as: “1-7”;

2) Line 29: the authors provide one short sentence and cite 7 works. Then, in the next sentence, the authors write that nanocellulose is well suited for these purposes (increasing the strength of paper). Then the question is: what do these 7 works suggest to increase the strength of paper? Please describe briefly.

3) Table 2: What causes the reduction in breaking length to 0.37 km compared to the control (1.85 km) for unbleached kraft pulp K=49.7?

4) Why did you limit yourself to adding 5% nanocellulose? In their conclusions, the authors indicate that less than 3% is impractical. But do they not consider what effect nanocellulose above 5% will have on the strength of paper?

Author Response

Dear Reviewer,
we have tried to correct the manuscript according to your recommendations.
We hope that everything will be clearer now.

Responses to your comments are attached.

Sincerely,
Authors

Reviewer 2 Report

Comments and Suggestions for Authors

In page 1 in introduction section, you give the examples of coating materials such as some polymers, waxes, PE, PP, PTFE or different polyesters. PE, PP, PTFE and polyesters belong to the polymers! Pleaste correct it.

The method of spraying and coating applied in the study should be described in detail.

In page 4 it is unclear why you decided to apply flax as you have previously cooked pulps. Moreover, in Table 2 you give pulps of Kappa number from 19.2 to 49.7, so how have you decided to use pulp cooked to Kappa number of 19.9?

It is not clear why you have added 5% of NC since you have stated 0.5% of NC to be sufficient.

In page 8, according to presented bibliography, did authors apply NC?

In Figure 2, it is unclear what was the aim to present it? Are the dots the same? If not, please give the legend and increase the points.

Comments on the Quality of English Language

The quality of the English Language should be highly improved. Some sentences are to long and not understandable.

The tense should be uniform in all the text.

Author Response

Reviewer No. 2

Dear Reviewer,

we have tried to correct the manuscript according to your recommendations.

We hope that everything will be more apparent now.

Responses to your comments are attached.

Sincerely,

Authors

Reviewer 3 Report

Comments and Suggestions for Authors

The manuscript can be accepted after carrying out these corrections.

- What is the novelty of the case of study?

- In Experiment part section two, the additions of TEMPO-Oxidized as coating is not clear. How can you use 2.5g CNF per hand sheet and also 3.5, 4.5 gm CNF (Too much amount)?

- What is the retain amount of CNF in the hand sheet after paper making in both cases (additions and coating)?

- What is the economic impact for this study?

Comments on the Quality of English Language

The manuscript needs to rewrite especially there are a lot of words are separated and they must be one word e.g. word properties in line 43, word readded in line 47, line 86, 108.

Author Response

Reviewer No. 3,

Dear Reviewer,

we have tried to correct the manuscript according to your recommendations.

We hope that everything will be apparent now.

Responses to your comments are attached.

Sincerely,

Authors

Reviewer 4 Report

Comments and Suggestions for Authors

Even though the Introduction contains relatively recent citations, it does not provide compelling evidence to support the need to pursue this research and fails to define the specific focus of the research and research gaps to address. The Introduction does not provide the context of using nanocellulose in papermaking – for example, what types of nanocellulose are considered for application in the wet-end and in the surface treatment phases of paper production; what addition levels are considered; what are the challenges, especially regarding the retention of nanocellulose used in the wet-end of papermaking. What are the current practices in the application of nanocellulose materials in paper production? In the Materials and Methods, authors introduce two types of nanocellulose that they used in the paper production – they fail to explain why these two types of nanocellulose were used, how are they different from each other, are there any reasons to expect that one would provide advantages over the other either in the wet-end phase or in the surface treatment phase of the paper production?

The Materials and Methods section should be improved. For example, line 71 reports on the delignification degree of kraft unbleached pulps, whereas unbleached pulps should be identified by their kappa number; bleached pulp is identified by its kappa number, 19.2. This kappa number corresponds to ~2.9% residual lignin (Lignin content, % = Kappa number *0.15;  T 236 om-22) that is in the range of bleachable grades for hardwood pulps (Hart (2011) Tappi J. 10:37-41).

Line 78 and Table 1: Supermass colloid?

Lines 75&97 and Table 1: TEMPO Oxided?

What was the retention of NC in the production of paper when NC was added to the pulp slurry? Only in the Conclusions do the authors acknowledge that the retention of NC should be  addressed “In general, it can also be observed that additions of less than 3 % do not significantly 287 affect the resulting properties, as apparently the nanocellulose fibrils are introduced into the sub-sieve waters of the paper machine.” While they fail to discuss this issue throughout the main body of the paper.

How was coating performed? How was spraying performed? How was the uniformity of spraying/coating controlled? It is very important that these procedures are explained in detail as later in the text, these two procedures show different results and it is concluded that Lines 284-285  “spraying nanocellulose on the paper appears to be the most effective.”

How many samples of the same formulation were made?

The statistical method used in the analysis of the results (Duncan’s test) should be described in the Materials and Methods.

Why was the pre-experiment with kraft pulp performed?  How was the decision to use the flax soda pulp of kappa number 19.9 in future experiments made based on the experiments performed with kraft pulp? Specifically, if the pre-experiments revealed that the addition of nanocellulose provides the most beneficial effects on the paper produced from the pulp of the lowest kappa number, why the second series of experiments did not include this pulp?

The manuscript reports strength properties that are related to each other - specifically Breaking length and Tensile index [the relationship is listed in ref #29] and those which are closely related to each other - burst strength and tensile strength - without explanation of the significance of these parameters and the effect which NC may have on these paper parameters.

Line 136 “soda pulp” vs line 137 “a natron flax pulp” Was the soda or natron pulp used in these experiments?

Pre-experiments were performed with the addition of 0.5-1-3% of NC whereas experiments with natron flax or soda pulp (not clear which one) were performed with the addition of 0.5-1-3 and 5% of NC. What results from pre-experiments indicated a need to increase the level of NC addition further?

Authors report the tensile index and breaking length results and discuss them as they are independent parameters – Table 2, Figure 1, and for example lines 172-174. [ref #29 defines the relationship]

The results are listed as simple trends (increase/decrease) and without the discussion of potential factors leading to these trends.

Line 225 –define  “market length;” Line 227 – define “the marketable length I”

Lines 226 -229 “Compared to the work of Rahman and Uddin [29], the result for 226 cotton pulp alone was 2.5 km. They tried to increase the marketable length I by increasing 227 the lignin and Caro's acid content. They achieved values around 3.5 km by adding 6% 228 Caro's acid to the bagasse pulp [30].” Unclear who was using the increase in lignin content and Caro’s acid to improve “marketable length” – research group cited as [29]/[30].

Lines 285-286 “At 3 and 5 % NC content, the properties are comparable to papers using more NC but applied differently.” What higher levels of NC were used?

Only in the Conclusions do the authors acknowledge that the retention of NC should be addressed. “In general, it can also be observed that additions of less than 3 % do not significantly 287 affect the resulting properties, as apparently the nanocellulose fibrils are introduced into 288 the sub-sieve waters of the paper machine.

Comments on the Quality of English Language

English editing would be helpful 

Author Response

Dear reviewer,
we have attempted to revise the manuscript according to your recommendations.
We hope that everything will be clear now.

The comments are attached.

Sincerely,
Authors
